# Compliance with the 24-Hour Movement Behavior Guidelines and Associations with Adiposity in European Preschoolers: Results from the ToyBox-Study

**DOI:** 10.3390/ijerph18147499

**Published:** 2021-07-14

**Authors:** Marga Decraene, Vera Verbestel, Greet Cardon, Violeta Iotova, Berthold Koletzko, Luis A. Moreno, María L. Miguel-Berges, Beata Gurzkowska, Odysseas Androutsos, Yannis Manios, Marieke De Craemer

**Affiliations:** 1Department of Rehabilitation Sciences, Ghent University, 9000 Ghent, Belgium; marga.decraene@ugent.be (M.D.); vera.verbestel@ugent.be (V.V.); 2Department of Movement and Sports Sciences, Ghent University, 9000 Ghent, Belgium; greet.cardon@ugent.be; 3Department of Pediatrics, Medical University of Varna, UMHAT ‘St. Marina’, 9002 Varna, Bulgaria; iotova_v@yahoo.com; 4Dr. von Hauner Children’s Hospital, University of Munich Medical Centre, 80337 München, Germany; Berthold.koletzko@med.uni-muenchen.de; 5GENUD (Growth, Exercise, NUtrition and Development) Research Group, University of Zaragoza: Instituto Agroalimentario de Aragón (IA2), Instituto de Investigación Sanitaria Aragón (IIS Aragón), 500001 Zaragoza, Spain; lmoreno@unizar.es (L.A.M.); mlmiguel@unizar.es (M.L.M.-B.); 6Centro de Investigación Biomédica en Red de Fisiopatología de la Obesidad y Nurtrición (CIBEROBN), Instituto de Salud Carlos III, 28029 Madrid, Spain; 7Public Health Division, The Children’s Memorial Health Institute, 04-730 Warsaw, Poland; B.Gurzkowska@IPCZD.PL; 8Department of Nutrition and Dietetics, University of Thessaly, 421 00 Trikala, Greece; oandroutsos@uth.gr; 9Department of Nutrition and Dietetics, Harokopio University, 177 78 Athens, Greece; manios.toybox@hua.gr; 10Research Foundation Flanders, 1000 Brussels, Belgium

**Keywords:** 24 h guidelines, movement behavior, physical activity, sedentary behavior, screen time, sleep, compliance, adiposity, preschooler

## Abstract

In 2019, the World Health Organization (WHO) published 24 h movement behavior guidelines for preschoolers with recommendations for physical activity (PA), screen time (ST), and sleep. The present study investigated the proportion of preschoolers complying with these guidelines (on a total week, weekdays and weekend days), and the associations with adiposity. This cross-sectional study included 2468 preschoolers (mean age: 4.75 years; 41.9% boys) from six European countries. The associations were investigated in the total sample and in girls and boys separately. PA was objectively assessed by step counts/day. Parent-reported questionnaires provided ST and sleep duration data. Generalized estimating equations were used to analyze the association between guideline compliance and adiposity indicators, i.e., body mass index (BMI) z-score and waist to height ratio (WHR). Only 10.1% of the preschoolers complied with the 24 h movement behavior guidelines, 69.2% with the sleep duration guideline, 39.8% with the ST guideline and 32.7% with the PA guideline. No association was found between guideline compliance with all three movement behaviors and adiposity. However, associations were found for isolated weekday screen time (BMI z-scores and WHR: *p* = 0.04) and weekend day sleep duration (BMI z-scores and WHR: *p* = 0.03) guideline compliance with both lower adiposity indicators. The latter association for sleep duration was also found in girls separately (BMI z-scores: *p* = 0.02; WHR: *p* = 0.03), but not in boys. Longitudinal studies, including intervention studies, are needed to increase preschoolers’ guideline compliance and to gain more insight into the manifestation of adiposity in children and its association with 24 h movement behaviors from a young age onwards.

## 1. Introduction

Overweight (OW) and obesity (OB) among preschoolers (3–5 years of age) are major public health concerns. In 2019, the World Health Organization (WHO) estimated that 38 million children under five years of age had OW or OB [1]. This is particularly problematic as these young children are at risk for being obese in later childhood and even adulthood [2,3]. Moreover, having OW or OB is associated with the risk for the development of a wide range of physical and psychological co-morbidities [4,5], including noncommunicable diseases, such as cancer, cardiovascular disease and diabetes mellitus type 2, later in life [1]. A sufficient amount of physical activity (PA), low levels of sedentary behavior (SB) and a sufficient amount of sleep are known to counteract OW and OB [6,7,8,9,10,11,12]. These behaviors can more easily be shaped in children under five years of age, as they are most susceptible for changes in behavioral habits [13]. Taking this together with the knowledge that habits track throughout life [14,15], the prevention of OW and OB from a young age is highly required.

In the past, studies targeted PA, SB and sleep in isolation [6,7,9]. These behaviors are, however, part of a continuum of movement across one 24 h day: changes in the time spent on one behavior evidently change the duration of at least one other behavior [16,17]. Moreover, a recent review suggested that specific combinations of these behaviors (high PA, low SB, and high sleep) are associated with favorable health indicators for children under five years of age, including adiposity indicators [18]. Evidence was, however, limited due to the small number of studies [18]. Additionally, the WHO acknowledges the importance of targeting PA, SB and sleep in an integrated approach to prevent OW and OB. This is reflected in the release of 24 h movement behavior guidelines for children under five years of age [19]. According to these guidelines, in preschoolers, a healthy 24 h day should include: (1) at least 180 min of PA of which at least 60 min is moderate to vigorous PA per day, (2) no more than 60 min of sedentary screen time per day, and (3) a good quality sleep of 10–13 h per day [19].

Several studies examined the proportion of preschoolers complying with the 24 h movement behavior guidelines [20,21,22,23,24,25,26,27]. To our knowledge only four of these studies investigated the association between preschoolers’ compliance with 24 h movement behavior guidelines and adiposity indicators [22,23,24,25]. At first, a Canadian study found no associations between compliance with the 24 h movement guidelines and weight status (“normal weight” versus “being at risk for OW”, and “being OW or obese”), or for body mass index (BMI) z-scores [22]. A Finnish and a Chinese study reported no associations between compliance with the 24 h movement behavior guidelines and adiposity indicators (BMI, waist circumference (WC) and weight status (normal weight versus OW/obese)) [23,24]. Although no associations were found for the compliance with all three behaviors, associations were found for the compliance with two out of three behaviors (i.e., PA and sleep) and lower BMI and WC (BMI: B −0.30, 95% confidence interval (CI) −0.52 to −0.08; WC: B −1.03, 95% CI −1.53 to −0.52) in the Finnish study [23]. On the other hand, within the Chinese study, preschoolers who did not comply with the screen time guideline had a higher risk for OW or OB (Odds Ratio (OR) 3.76, 95% CI 1.50–9.45) [24]. Lastly, a Japanese study found an association between compliance with all three movement behavior guidelines and weight status (“normal weight” versus “OW/obese”). Preschoolers who did not comply with all three movement behavior guidelines were more likely to have OW or OB (OR 1.139, 95% CI 1.009 to 1.285) [25]. As the number of studies is limited and the already existing studies show rather mixed results, it is important to further investigate the association between preschoolers’ compliance with the 24 h movement behavior guidelines and adiposity indicators. Furthermore, to our knowledge there are no studies examining preschoolers’ 24 h movement behavior guideline compliance and its associations with adiposity in a cross-national European sample. By investigating a cross-national sample, country-specific traits can be bypassed, such as policies, regulations, legislations and cultural traits. Therefore, the results become more generalizable and can apply to a broader population of different nationalities. As the WHO 24 h movement behavior guidelines were created for a global population and aim to be “general” [19], they should also be studied in a more generalizable study sample. Therefore, the present study aimed to investigate the proportion of preschoolers complying with the 24 h movement behavior guidelines and the associations between this compliance and adiposity indicators among a European cross-national sample. In addition, previous studies show differences in PA, SB or screen time and sleep duration on weekdays versus weekend days and for boys versus girls [28,29,30,31]. Additionally, adiposity indicators seem to be different in boys and girls [28]. However, no studies have investigated the association between 24 h movement behavior guidelines and adiposity indicators in boys and girls and on weekdays and weekend days separately. Therefore, this was an additional aim of the present study.

## 2. Materials and Methods

### 2.1. Study Protocol

Subjects in the present study participated in the baseline measurements of the European ToyBox-study (www.toybox-study.eu, accessed on 15 March 2021). A more detailed description of the ToyBox-study can be found in previous publications [32,33]. The aim of the ToyBox project was to prevent OW and OB in four- to six-year-old preschoolers by means of a multidisciplinary intervention in six European countries (Belgium, Bulgaria, Germany, Greece, Poland, and Spain). For each of these countries, participants were recruited at kindergartens, daycare centers, or preschool settings. In this study, these settings are referred to as “kindergartens”. The kindergartens were selected from one or two provinces per country. Within these provinces, information on the socio-economic status (SES) was obtained for each municipality (years of education for the population of 25–55 years (cut-off: >14 years of education) or annual income (quantitative variable)). The municipalities were divided into low, medium and high SES municipalities using tertiles. For each SES category, five municipalities were randomly selected. Within those municipalities, all the kindergartens were listed and the 20% smallest kindergartens were excluded based on the lowest number of preschoolers. The other 80% of kindergartens were invited to participate, and those that agreed to participate were included in the study. The ToyBox-study aimed to recruit a minimum of 1100 preschoolers and one parent/caregiver per preschooler for each country. This minimum number of preschoolers within each country was based on power analyses and a previous school-based intervention [34]. All parents/caregivers of preschoolers born in 2007 and 2008 received an information letter about the study. The children whose parents/caregivers provided written informed consent participated in the study. Data were collected in May and June 2012. This was springtime in all participating countries. In total, 7854 preschoolers participated in the baseline measurements and had valid data on one or more variables.

Before the study-onset, the study was approved in line with national regulations by Ethical Committees in all six countries (i.e., the Ethical Committee of Ghent University Hospital (Belgium), Committee for the Ethics of the Scientific Studies (KENI) at the Medical University of Varna (Bulgaria), Ethikkommission der Ludwig- Maximilians-Universität München (Germany), the Ethics Committee of Harokopio University of Athens (Greece), Bioethics Committee of Children’s Memorial Health Institute (Poland), and CEICA (Comité Ético de Investigación Clínica de Aragón (Spain)).

### 2.2. Measurements

The measurements and the procedure of data collection, data deposition and data reporting were standardized within the ToyBox-study [33]. Body weight, height and WC of participating preschoolers were measured to assess adiposity. Participating preschoolers were fitted with a motion sensor to assess their PA levels. In addition, parents/caregivers completed a parental questionnaire (Principal Caregiver’s Questionnaire) on socio-demographics, health related behaviors including PA, ST and sleep duration, and determinants of these behaviors. The parents/caregivers who completed this questionnaire are the principal caregivers of the participating preschoolers. The Principal Caregiver’s Questionnaire can be found online (www.toybox-study.eu). Detailed information on the development and reliability of this questionnaire was described in previous publications [33,35].

#### 2.2.1. Adiposity

Body weight, height and WC were measured by trained research assistants at kindergartens according to standardized protocols [33]. A detailed description regarding the procedures and training of research staff and validity and reliability of the measurements can be found in a previous publication [36]. Preschoolers were measured with bare feet, while wearing light clothing. Body height was measured to the nearest millimeter by means of the SECA 225 or SECA 214 Leicester Portable stadiometer (Seca, Hamburg, Germany). Body weight was measured to the nearest 0.1 kg by a calibrated electronic scale SECA 861 or SECA 813. WC was measured to the nearest 0.1 cm with a SECA 200 or SECA 201 unelastic tape, with the preschooler at an upright position. For all adiposity measurements, two readings were conducted and the mean was calculated for analyses. If the readings differed by more than 1%, a third measurement was obtained and the mean of the two measurements with the slightest difference was calculated. Weight and height were used to calculate BMI (kg/m^2^). Weight status was calculated in line with previous ToyBox-studies and BMI z-scores were calculated on the basis of the WHO Anthro and AnthroPlus software [37]. Sex- and age-specific BMI z-scores provide a relative measure of adiposity adjusted for age and sex [38]. The z-score is the number of standard units that a person’s BMI deviates from a mean or reference value. WC and height were used to calculate the waist-to-height ratio (WHR).

#### 2.2.2. 24 h Movement Behaviors

##### Physical Activity

PA was assessed by means of the daily number of steps using Omron Walking Style Pro pedometers (HJ-720IT-E2, Omron, Kyoto, Japan) (Bulgaria, Germany, Greece, Poland, and Spain) and GT1M, GT3X and GT3X+ ActiGraph (Pensacola, FL, USA) accelerometers (Belgium). Both Omron Walking Style Pro pedometers and Actigraph accelerometers were found to be valid for measuring step counts in preschoolers [39]. To make sure the devices were worn correctly, preschoolers’ parents/caregivers received an informational letter with instructions. Parents/caregivers were asked to let their preschooler wear the motion sensor for six consecutive days during all waking hours, including two weekend days and to remove it only during water-based activities. The sensors were worn on the right hip and secured by an elastic waist band. After the data collection, data from the pedometers were downloaded using Omron Health Management Software version E1.012 (Omron, Kyoto, Japan), and data from accelerometers were downloaded using ActiLife version 5.5.5-software (ActiGraph, Pensacola, FL, USA). Data of the first (fitting day) and sixth day (collection day) were omitted, as these days had incomplete data. All daily step counts below 1000 and above 30,000 steps were considered outliers and therefore recoded into missing data [40]. Daily step counts were separately calculated for weekdays and weekend days. Average daily step counts for a total week were calculated as follows: ((mean daily step counts on weekdays) × 5) + (mean daily step counts on weekend days) × 2)/7. Preschoolers’ step count data were only included in data analyses when there were valid data for at least two weekdays and one weekend day. The number of steps per day was dichotomized into 0 (<11,500 steps/day) and 1 (≥11,500 steps/day), to calculate the percentage of preschoolers complying with the PA guideline of 11,500 steps/day which corresponds to 180 min of total PA per day [41].

##### Screen Time

Screen time was assessed by questioning television viewing and computer use separately in the Principal Caregiver’s Questionnaire. The question regarding television viewing was formulated as follows: “About how many hours a day does your child usually watch television (including DVDs and videos) in his/her free time?”. Possible answers were “never”, “less than 30 min/day”, “30 min to <1 h/day”, “1–2 h/day”, “3–4 h/day”, “5–6 h/day”, “7–8 h/day”, “8 h per day”, “more than 8 h/day”, and “I don’t know”. For computer use, the question was as follows: “About how many hours a day does your child use the computer for activities such as playing games on a computer, game consoles (e.g., PlayStation, Xbox, GameCube) during leisure time?”. The answer possibilities were identical to the question about television viewing. The questions for both screen behaviors were asked separately for weekdays and weekend days. Possible answer categories were re-coded into minutes of television viewing and computer use per day by means of the midpoint method [30]: “less than 30 min/day” into “15 min/day”, “30 min to <1 h/day” into “45 min/day”, “1–2 h/day” into “90 min/day”, “3–4 h/day” into “210 min/day”, “5–6 h/day” into “330 min/day”, “7–8 h/day” into “450 min/day”, “8 h per day” into “480 min/day”, and “more than 8 h/day” into “540 min/day” Daily minutes of television viewing and computer use were added up to reflect total screen time on weekdays and weekend days. Overall screen time was calculated as follows: ((screen time on weekdays) × 5) + (screen time on weekend days) × 2)/7. To calculate the percentage of preschoolers complying with the screen time recommendation, minutes of total screen time were dichotomized into 0 (>60 min of screen time per day) and 1 (≤60 min of screen time per day).

##### Sleep Duration

Sleep duration was assessed by the Principal Caregiver’s Questionnaire. The following question was asked for both weekdays and weekend days separately: “How many hours does your child usually sleep during the night.” Possible answers were “less than 6 h”, “6–7 h”, “8–9 h”, “10–11 h”, “12–13 h”, “14 h”, “more than 14 h”, and “I don’t know”. Similar to the screen time items, possible answer categories were re-coded into minutes of sleep by means of the midpoint method [30]: “less than 6 h” into “5 h”, “6–7 h” into “6.5 h”, “8–9 h” into “8.5 h”, “10–11 h” into “10.5 h”, “12–13 h” into “12.5 h”, “14 h” remained “14 h”, and “more than 14 h” was re-coded into “15 h” Overall sleep duration was calculated as follows: ((sleep duration on weekdays) × 5) + (sleep duration on weekend days) × 2)/7. To calculate the percentage of preschoolers complying with the sleep duration recommendation, the answers “10–11 h” and “12–13 h” were recoded into 1. The other answer possibilities were recoded into 0, reflecting a shorter or longer sleep duration than recommended.

#### 2.2.3. Covariates

Preschoolers’ sex and age and the educational level of the parents/caregivers who completed the questionnaire were included as covariates and obtained through the Principal Caregiver’s Questionnaire. The parents/caregivers were asked to report the date of birth and sex of their child. The educational level of the parent/caregiver was assessed by the number of years of education and used as a proxy for family SES [42,43,44]. Possible answers were “less than 7 years” “7–12 years”, “13–14 years”, “15–16 years” and “more than 16 years”. For ease of interpretation, educational level was dichotomized into low (14 or fewer years of education) and high (more than 14 years of education) SES.

### 2.3. Data Analysis

SPSS (version 26) was used for data analysis. Continuous data were checked and found to be normally distributed (skewness < 0.70). Descriptive characteristics of the study sample are presented as means and standard deviations for continuous variables and as percentages for categorical variables. Descriptives are provided for the total study sample, and for boys and girls separately. The average time spent on PA, screen time and sleep duration are provided for the total week, for weekdays and for weekend days. The proportion of preschoolers complying with the integrated 24 h movement guidelines are presented in Venn diagrams. To analyze the associations between compliance with the 24 h movement behavior guidelines and adiposity indicators (i.e., dependent variable; BMI z-scores and WHR), generalized estimating equation (GEE) models were used with two levels (class; kindergarten). This study aimed to present generalized data. Therefore, country (6 categories) was not included as a cluster level. However, it was included as a covariate, together with age and sex of the preschooler and educational level of the parent/caregiver who completed the Principal Caregiver’s Questionnaire. The associations between compliance with the 24 h movement behavior guidelines and the adiposity indicators were analyzed for the overall week and separately for weekdays and weekend days. The GEE models were conducted a second time to check for interaction effects between sex and compliance with the 24 h movement behavior guidelines. In the case of significant interaction effects, the association between guideline compliance and adiposity for boys and girls was analyzed separately. Subsequently, the GEE models were repeated separately for boys and girls. Attrition analyses were performed to compare preschoolers with valid data to preschoolers without valid data. For this, a logistic regression analysis was performed with two levels (class; kindergarten). For all analyses, statistical significance was set at *p* < 0.05.

## 3. Results

In total, parents/caregivers of 7854 preschoolers from six European countries completed the Principal Caregiver’s Questionnaire. Of those preschoolers, 4555 wore a motion sensor to measure PA (i.e., step counts). A total of 2669 children had valid data for PA, SB and sleep for both weekdays and weekend days. Of those preschoolers, 2468 also had valid data for weight, height and the covariates (31.4% of total sample and 54.2% of the sample that received a motion sensor to measure PA). These 2468 preschoolers represent the study sample of the current study. For analyses with WHR, four more preschoolers were excluded as they had no WC data. Attrition analyses showed that preschoolers with a parent/caregiver with a lower educational level were more likely to have incomplete data compared to preschoolers with a parent/caregiver with a higher educational level (OR = 1.52; 95% CI = 1.38–1.68); no differences were found for sex (OR = 0.95; 95% CI = 0.86–1.05); and for age (OR =0.93; 95% CI = 0.84–1.02).

Table 1 shows the descriptive characteristics of the total sample (*n* = 2468) included in the current study and stratified by sex. In total, 12.8% of the study sample had OW or OB. The average step counts were 10,485 steps/day (SD = 2866), whereas the average screen time was 92.76 min/day (SD = 67.12). Furthermore, preschoolers slept on average 10.20 h/night (SD = 1.06). Step counts, screen time and sleep duration were significantly different for weekdays versus weekend days (*p* < 0.001). Boys had a higher BMI z-score and WC, more step counts, more screen time, and a shorter sleep duration on weekend days compared to girls (*p* < 0.05).

Figure 1 shows the compliance of the current study sample with the 24 h movement behavior guidelines overall (i.e., total week), on weekdays and on weekend days, including the proportion of preschoolers complying with the separate guidelines for physical activity, screen time and sleep duration, and combinations of these guidelines. Overall, 10.1% of the preschoolers complied with the 24 h movement behavior guidelines. On weekdays, the proportion of preschoolers complying with the guidelines was 16.8%, whereas 6.8% complied with the guidelines on weekend days. In the current study sample, 3.6% (*n* = 88) complied with the 24 h movement behavior guidelines on both weekdays and weekend days. When focusing on the separate recommendations, the sleep duration guideline had the highest compliance rate (69.2% overall, 73.4% on weekdays, 81.9% on weekend days), followed by the screen time guideline (39.8% overall, 55.6% on weekdays, 25.4% on weekend days). The physical activity guideline had the lowest compliance rate (32.7% overall, 37.6% on weekdays, 28.7% on weekend days). In total, 15.0% of the preschoolers did not comply with any of the guidelines overall; 10.2% on weekdays and 10.6% on weekend days. Preschoolers’ compliance with the 24 h movement behavior guidelines for each participating country separately can be found in Appendix A. Preschoolers’ guideline compliance within the participating countries range from 0% (Bulgaria) to 27.9% (Germany) overall, 0% (Bulgaria) to 36.7% (Germany) on weekdays and 1.5% (Bulgaria) to 18.1% (Germany) on weekend days.

Table 2 shows the associations of compliance with the 24 h movement behavior guidelines with BMI z-score on the one hand, and WHR on the other hand, for separate guidelines for physical activity, screen time and sleep duration, and combinations of these guidelines. The mean BMI z-score and WHR of preschoolers complying and not complying with the guidelines are also presented in this table. No associations were found between preschoolers’ compliance with the combination of all three guidelines and BMI z-score or WHR. Preschoolers had significantly lower BMI z-score and WHR when complying with the guideline for screen time on weekdays (BMI z-score *p* = 0.04; WHR *p* = 0.04) and complying with the guideline for sleep duration on weekend days (BMI z-score *p* = 0.03; WHR *p* = 0.03). BMI z-score was also significantly lower when preschoolers complied with the combination of guidelines for screen time and sleep duration overall and on weekdays (*p* = 0.01). Some models showed a significant interaction between sex and compliance with the 24 h movement behavior guidelines. This confirms the relevance of analyzing the association between the guidelines and adiposity for boys and girls separately.

Table 3 shows the associations of compliance with the 24 h movement behavior guidelines with BMI z-score and WHR in boys. The mean BMI z-score and WHR of boys complying and not complying with the guidelines are also presented. For compliance with all three guidelines, no associations with BMI z-score or WHR were found. On the other hand, boys had a significantly lower BMI z-score when complying with the guideline for screen time on weekend days (*p* = 0.02), and with the combination of guidelines for screen time and sleep duration on weekend days (*p* = 0.03). Significantly lower WHR in boys was found when complying with the guideline for physical activity on weekend days (*p* = 0.02) and when complying with the combination of guidelines for physical activity and sleep duration on weekend days (*p* = 0.04). Table 4 shows the associations of compliance with the 24 h movement behavior guidelines with BMI z-score and WHR in girls. The mean BMI z-score and WHR of girls complying and not complying with the guidelines are also presented. There were no associations between compliance with all three guidelines and BMI z-score or WHR in girls. When complying with the guideline for sleep duration on weekend days, girls showed a significantly lower BMI z-score (*p* = 0.02) and lower WHR (*p* = 0.03). Additionally, a significantly higher BMI z-score was found when girls complied with the guideline for physical activity in a total week (*p* = 0.02) and on weekend days (*p* = 0.01).

## 4. Discussion

This is the first study investigating the compliance with the 24 h movement behavior guidelines and associations with adiposity indicators among preschoolers of a cross-national European sample. This is also the first study investigating this association on weekdays and weekend days in boys and girls separately. The overall proportion of preschoolers complying with the 24 h movement behavior guidelines was small (10.1%). This is in line with previous studies reporting guideline compliance in preschoolers ranging from 4.5 to 23.6% [21,22,23,24,25,26,27]. As all of these studies show low guideline compliance, better and scalable interventions are needed to optimize preschoolers’ 24 h movement behaviors. In addition, the proportion range of preschoolers’ guideline compliance in previous studies is wide. When investigating the countries separately, the current study found guideline compliance ranging from 0% (Bulgaria) and 27.9% (Germany). This might be explained by country-specific differences in policies, legislation, regulations and culture with regard to PA, screen time and sleep. At the moment, we do not have clear insights into these differences and how they impact the behaviors. For instance, the participating countries, except for Greece, organize physical education classes at kindergartens, but for most countries it is unclear how much time is actually dedicated to these classes [45]. Future research should provide more insight. When investigating the proportion of preschoolers’ compliance with the 24 h movement behaviors on weekdays and weekend days separately, weekday compliance (16.8%) was higher than weekend day compliance (6.8%). The study of Leppänen et al. (2019) also investigated weekdays and weekend days separately and found similar results [23]. The difference between weekdays and weekend days might particularly be explained by the higher proportion of preschoolers complying with the screen time guideline on weekdays (55.6%) compared to weekend days (24.5%). The use of parental-reported screen time might imply an underestimation of preschoolers’ screen time on weekdays. As preschoolers attend kindergarten during the day, parents might not have insights in preschoolers’ screen time at the kindergarten. However, it is possible that screens are rarely used in these kindergartens. Therefore, parents’ lack of insight into screen time at the kindergarten might only minimally have influenced the results. Future research should provide more insight into how often screens are used at kindergartens, especially taking the rapid technical evolution into account, which has made screen activities increasingly accessible [46,47,48].

In contrast with the expectations, no association was found for compliance with the integrated 24 h movement behavior guidelines and adiposity indicators (i.e., both BMI z-score and WHR). This is in agreement with a Canadian and a Chinese study, which also did not find any association between complying with a combination of movement behavior guidelines and adiposity [22,24]. On the other hand, Japanese researchers found that preschoolers complying with all three 24 h movement behavior guidelines were less likely to have OW or OB [25]. A Finnish study found an association for the combination of PA and sleep duration guideline compliance with lower BMI and with lower WC [23]. Differences in study methodology (e.g., different cut-points used to measure PA) and the differences in the proportion of preschoolers complying with the guidelines are plausible explanations for these mixed findings between studies. Another plausible explanation might be that preschoolers are too young to observe the impact of health behaviors on adiposity. It is possible that the impact of an insufficient amount of time spent on each of the 24 h movement behaviors at a young age is manifested later in life. This is supported by previous studies showing significant associations between more favorable 24 h movement behaviors (in isolation and in combination) and lower adiposity in older children [8,11,49,50,51]. Longitudinal studies already starting at preschooler age might provide more accurate insights into the manifestation of adiposity and its association with 24 h movement behaviors over time.

While no association with both BMI z-score and WHR was found for integrated guideline compliance, some associations were found with isolated guideline compliance. Isolated weekday screen time and weekend day sleep duration guideline compliance were associated with lower BMI z-score and WHR. At first, this highlights the need to decrease screen time on weekdays. The association found within the current study regarding screen time is comparable with a Chinese study [24]. Chinese preschoolers who complied with the screen time guideline were less likely to have OW or OB. However, in the latter study, no distinction was made between week- and weekend days. In addition, no associations were found for compliance with any of the other guidelines or combinations of guidelines with adiposity [24]. The association between screen time and adiposity is supported by evidence in both preschoolers and older children. Higher screen time is associated with more unhealthy dietary intake, and this, in turn, leads to adiposity [52,53]. Secondly, the association between weekend day sleep duration guideline compliance and adiposity is in agreement with the Finnish study [23]. However, the latter association was found for both the total week and for weekend days, whereas the current study only found an association for weekend days. Although already high compliance rates were found for sleep duration, this does not alter the fact that both studies show the need to increase compliance to the sleep duration guideline on weekend days to prevent adiposity.

When considering boys and girls separately, there were no associations for compliance with the integrated 24 h movement behavior guidelines with BMI z-score and WHR. Only one other study investigated the association between 24 h movement behavior guideline compliance and adiposity in boys and girls separately. This Canadian study did not find any associations between integrated or isolated guideline compliance and adiposity [22]. The current study, however, showed an association between isolated weekend day sleep duration guideline compliance and lower BMI z-scores and WHR in girls, but not in boys. Other studies investigated the association between sleep duration in isolation and adiposity in preschoolers and older children [54,55,56,57]. They all showed an association for shorter sleep duration with higher adiposity. Not having an appropriate amount of sleep is suggested to increase hunger and appetite for high caloric foods and, in turn, leads to adiposity [58]. This supports the association between sleep duration guideline compliance and adiposity. However, in contrast to our results, the studies showed either no differences between sexes or only showed this association in boys [54,55,56,57]. Given the limited and mixed evidence at this time, further research is needed to investigate differential associations between boys and girls.

Strengths of the current study were the inclusion of a relatively large sample of preschoolers from six different European countries and the use of a standardized measurement protocol in all these countries which provides more generalizable data. Moreover, the current study was the first to investigate the association between compliance with the 24 h movement behavior guidelines and adiposity indicators on weekdays and weekend days in boys and girls separately. This provides insight into which moments of the week (weekdays versus weekend days) and which groups (girls versus boys) are most susceptible for future intervention studies.

The current study also had some limitations. First, the sample size of each participating country was not equal. Consequently, some countries weigh more with regard to preschoolers’ (lack of) compliance with the 24 h movement behavior guidelines. When associating guideline compliance with adiposity indicators, the analyses were, however, adjusted for country. In addition, preschoolers with a lower educated parent/caregiver were more likely to have incomplete data compared to preschoolers with a higher educated parent/caregiver, which may hamper the generalizability of our findings. People with a lower SES are a hard to reach population that should receive special attention in future research [59,60]. Other limitations of the current study were the use of step counts to measure PA and the use of proxy-reported screen time and sleep duration. By using step counts, we might have missed relevant PA data as we did not have information on the full PA guidelines (180 min of PA of which at least 60 min moderate to vigorous PA per day). However, using pedometers is an adequate validated alternative to measure PA in large study samples as these monitors have a lower cost compared to accelerometers [39,40]. Using parental-reported questionnaires may have led to bias and over- or underestimation of the screen time and sleep duration data because of parents/caregivers’ social desirability and recall difficulties [61]. Another limitation was the small proportion of preschoolers complying with the 24 h movement behavior guidelines. Due to this small proportion, the association between guideline compliance and adiposity might have been statistically biased, especially for PA guideline compliance. This might explain the association between PA guideline compliance and a higher BMI z-score in girls, which was unexpected and not supported by previous research. Furthermore, this study was not able to provide causal associations between compliance with the 24 h movement behavior guidelines and adiposity indicators due to its cross-sectional design. Additionally, the data collection was completed in May and June 2012. Therefore, the results were only valid for data collected during spring. Furthermore, due to the technological evolution, the offer of screen activities has evolved rapidly in the last decade. Future research should interpret the screen time results with caution when making contemporary comparisons.

Based on these limitations, we suggest future research to use an objective instrument that is most accurate to meet the research questions of their study. For instance, to objectively measure all 24 h movement behaviors together, accelerometers could be used [23,24]. This would mean that the limitations of step count monitors and reporter bias can be avoided. These accelerometers should also be able to capture the intermittent patterns of preschoolers’ movement, which can be carried out by, for instance, tri-axial accelerometers such as the Actigraph GT3X+ [62]. The Actigraph GT3X+ is able to differentiate PA, SB and sleep [63]. However, the accuracy of measuring all 24 h movement behavior was not yet validated among preschoolers. Diaries can complement the objective data to obtain information about the context of the behaviors (e.g., screen time). Diaries are able to overcome recall difficulties for parents regarding their preschooler’s behaviors [61]. To investigate causal associations between compliance with the 24 h movement behavior guidelines and adiposity indicators, longitudinal studies, including intervention studies, are recommended.

## 5. Conclusions

Overall, low compliance with the 24 h movement behavior guidelines was found among a cross-national European sample of preschoolers. Furthermore, no association was found between the integrated 24 h movement behavior guidelines and adiposity indicators in the total study sample, nor separately in boys and girls on week- and weekend days. Intervention studies are needed to increase the proportion of preschoolers complying with the guidelines. To prevent OW and OB, it remains of great importance for preschoolers to adopt healthy lifestyle behaviors as specified by the 24 h movement behavior guidelines [19]. This is especially the case because children under five years of age can adopt new behavioral habits more easily [13], including lifestyle, and these habits track throughout life [14,15]. Longitudinal studies are required to provide more accurate insights in the manifestation of adiposity in children and its association with 24 h movement behaviors from a young age onwards.

## Figures and Tables

**Figure 1 ijerph-18-07499-f001:**
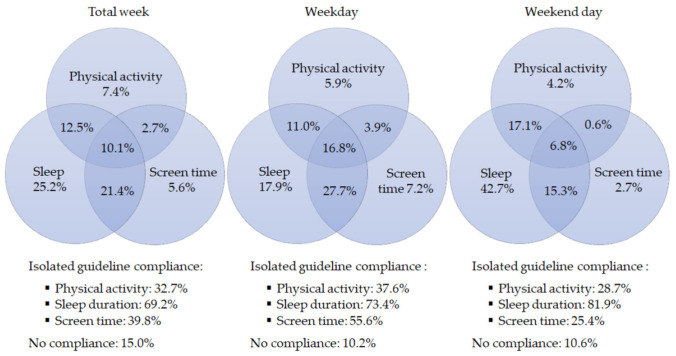
Venn diagrams showing the percentage of preschool children (not) complying with the 24 h movement behavior guidelines in a total week, on weekdays, and on weekend days. The sum of each circle is the equivalent to the percentage of preschool children complying with the isolated guidelines (e.g., 32.7% for overall physical activity).

**Table 1 ijerph-18-07499-t001:** Descriptive characteristics of the study sample.

	Total Sample	Boys	Girls	
Total (*N*)	2468	1034 (41.9%)	1164 (58.1%)	
Belgium (*n*)	570	299 (52.5%)	271 (47.5%)	
Bulgaria (*n*)	68	34 (50.0%)	34 (50.0%)	
Germany (*n*)	226	123 (54.4%)	103 (45.6%)	
Greece (*n*)	349	174 (49.9%)	175 (50.1%)	
Poland (*n*)	829	447 (53.9%)	382 (46.1)	
Spain (*n*)	426	227 (53.3%)	199 (46.7%)	
				***p*** **-value**
Mean age in years (SD)	4.75 (0.43)	4.74 (0.43)	4.76 (0.43)	0.33 ^a^
Parental education (%)				
Low (<14 years)	34.3	33.6	35.1	0.42 ^b^
High (≥14 years)	65.7	66.4	64.9	
Mean BMI (kg/m^2^) (SD)	15.81 (1.44)	15.83 (1.40)	15.77 (1.49)	0.32 ^a^
Mean BMI z-score (SD)	0.32 (0.95)	0.37 (0.98)	0.27 (0.92)	0.01 ^a^
Weight status (%)				
Underweight	11.2	11.1	11.3	0.37 ^b^
Normal weight	76.0	76.5	75.2	
Overweight	10.2	10.2	10.2	
Obesity	2.6	2.1	3.2	
Mean WC in centimeters (SD; *n* = 2464)	52.36 (3.81)	52.56 (3.70)	52.13 (3.91)	0.01 ^a^
Mean height in meters (SD)	1.09 (0.05)	1.09 (0.05)	1.08 (0.05)	<0.001 ^a^
Mean WHR (centimeters/meters) (SD; *n* = 2464)	48.12 (03.17)	48.12 (2.96)	48.11 (3.39)	0.93 ^a^
Mean total PA (steps/day) (SD)	10,485 (2866)	11,038 (2936)	9865 (2653)	<0.001 ^a^
Weekday *	10,802 (3211)	11,388 (3312)	10,146 (2960)	<0.001 ^a^
Weekend day *	9691 (3872)	10,163 (4016)	9163 (3631)	<0.001 ^a^
Mean total screen time (min/day) (SD)	92.76 (67.12)	96.04 (67.48)	89.08 (66.55)	0.01 ^a^
Weekday *	75.60 (62.41)	78.07 (62.04)	72.82 (62.73)	0.04 ^a^
Weekend day *	135.67 (98.53)	140.97 (101.21)	129.73 (95.12)	0.01 ^a^
Mean total sleep duration (h/day) (SD)	10.20 (1.06)	10.17 (1.08)	10.22 (1.04)	0.17 ^a^
Weekday *	10.08 (1.14)	10.09 (1.12)	10.09 (1.12)	0.53 ^a^
Weekend day *	10.49 (1.23)	10.43 (1.24)	10.56 (1.21)	0.01 ^a^

To compare means between boys and girls (^a^) independent sample t-tests were used for quantitative variables and (^b^) chi^2^ tests were used for categorical variables; (*) to compare means between weekdays and weekend days paired sample t-tests were used, resulting in a *p*-value of <0.001 for all comparisons. Abbreviations: *n* = number, SD = standard deviation, BMI = body mass index, WHR = waist to height ratio.

**Table 2 ijerph-18-07499-t002:** Associations of compliance with the 24 h movement behaviors with BMI z-score and WHR.

Movement Behavior	Compliance/Non-Compliance	BMI z-Score *n* = 2468					WHR *n* = 2464				
		Mean (SE)	B	95% CI Lower	95% CI Upper	*p*-Value	Mean (SE)	B	95% CI Lower	95% CI Upper	*p*-Value
Physical activity											
Total week	No compliance	0.35 (0.03)	−0.07	−0.16	0.01	0.10	47.99 (0.16)	−0.09	−0.37	0.20	0.55
	Compliance	0.42 (0.05)					48.08 (0.20)				
Weekdays	No compliance	0.36 (0.03)	−0.04	−0.12	0.04	0.36	48.01 (0.17)	−0.03	−0.29	0.24	0.84
	Compliance	0.39 (0.04)					48.04 (0.18)				
Weekend days	No compliance	0.36 (0.03)	−0.01	−0.10	0.07	0.75	48.06 (0.16)	0.15	−0.11	0.41	0.26
	Compliance	0.38 (0.04)					47.91 (0.19)				
Screen time											
Total week	No compliance	0.40 (0.04)	0.09	0.00	0.17	0.05	48.08 (0.16)	0.16	−0.09	0.41	0.22
	Compliance	0.32 (0.04)					47.92 (0.18)				
Weekdays	No compliance	0.42 (0.04)	0.09	0.01	0.17	**0.04**	48.16 (0.17)	0.26	0.01	0.50	**0.04**
	Compliance	0.33 (0.04)					47.90 (0.17)				
Weekend days	No compliance	0.39 (0.03)	0.08	−0.01	0.16	0.08	48.05 (0.16)	0.14	−0.13	0.40	0.32
	Compliance	0.31 (0.04)					47.92 (0.19)				
Sleep											
Total week	No compliance	0.42 (0.04)	0.08	−0.01	0.17	0.08	48.15 (0.18)	0.12	−0.10	0.52	0.18
	Compliance	0.34 (0.03)					47.94 (0.17)				
Weekdays	No compliance	0.41 (0.04)	0.07	−0.03	0.16	0.17	48.12 (0.19)	0.15	−0.17	0.47	0.37
	Compliance	0.35 (0.03)					47.97 (0.17)				
Weekend days	No compliance	0.45 (0.05)	0.11	0.01	0.21	**0.03**	48.30 (0.20)	0.36	0.03	0.69	**0.03**
	Compliance	0.34 (0.03)					47.94 (0.16)				
Physical activity and screen time											
Total week	No compliance	0.37 (0.03)	0.02	−0.10	0.14	0.73	48.04 (0.16)	0.16	−0.19	0.50	0.37
	Compliance	0.35 (0.06)					47.88 (0.22)				
Weekdays	No compliance	0.37 (0.03)	0.03	−0.07	0.13	0.58	48.04 (0.16)	0.12	−0.15	0.38	0.39
	Compliance	0.34 (0.05)					47.93 (0.19)				
Weekend days	No compliance	0.37 (0.03)	0.10	−0.04	0.23	0.17	48.04 (0.16)	0.25	−0.17	0.66	0.24
	Compliance	0.28 (0.07)					47.79 (0.25)				
Physical activity and sleep											
Total week	No compliance	0.37 (0.03)	0.01	−0.09	0.11	0.85	48.06 (0.16)	0.22	−0.08	0.53	0.15
	Compliance	0.36 (0.05)					47.84 (0.21)				
Weekdays	No compliance	0.38 (0.03)	0.03	−0.06	0.13	0.50	48.08 (0.16)	0.24	−0.04	0.52	0.09
	Compliance	0.34 (0.05)					47.84 (0.19)				
Weekend days	No compliance	0.37 (0.03)	0.02	−0.07	0.10	0.69	48.07 (0.16)	0.24	−0.04	0.51	0.09
	Compliance	0.35 (0.04)					47.83 (0.20)				
Screen time and sleep											
Total week	No compliance	0.40 (0.03)	0.09	0.00	0.18	**0.04**	48.04 (0.16)	0.07	−0.20	0.34	0.62
	Compliance	0.30 (0.04)					47.97 (0.18)				
Weekdays	No compliance	0.41 (0.04)	0.10	0.02	0.19	**0.01**	48.10 (0.17)	0.189	−0.06	0.44	0.14
	Compliance	0.31 (0.04)					47.91 (0.17)				
Weekend days	No compliance	0.39 (0.03)	0.09	−0.00	0.18	0.05	48.05 (0.16)	0.16	−0.12	0.43	0.27
	Compliance	0.30 (0.05)					47.90 (0.19)				
Physical activity, screen time and sleep											
Total week	No compliance	0.38 (0.21)	0.09	−0.04	0.22	0.19	48.05 (0.16)	0.33	−0.04	0.69	0.08
	Compliance	0.29 (0.07)					47.73 (0.23)				
Weekdays	No compliance	0.38 (0.03)	0.07	−0.05	0.18	0.25	48.05 (0.16)	0.18	−0.11	0.47	0.23
	Compliance	0.31 (0.06)					47.87 (0.20)				
Weekend days	No compliance	0.37 (0.03)	0.09	−0.05	0.24	0.18	48.04 (0.16)	0.25	−0.19	0.69	0.27
	Compliance	0.28 (0.07)					47.79 (0.26)				

Compliance with the 24 h movement behavior guidelines and associations with adiposity in European preschoolers: results from the ToyBox-study. Abbreviations: BMI = body mass index, WHR = waist to height ratio, *n* = number, SE = standard error, CI = confidence interval. Significant associations are marked in bold.

**Table 3 ijerph-18-07499-t003:** Association of compliance with the 24 h movement behaviors with BMI z-score and WHR in boys.

Movement Behavior	Compliance/Non-Compliance	BMI z-Score, Boys *n* = 1304					WHR, Boys *n* = 1300				
		Mean (SE)	B	95% CI Lower	95% CI Upper	*p*-Value	Mean (SE)	B	95% CI Lower	95% CI Upper	*p*-Value
Physical activity											
Total week	No compliance	0.43 (0.04)	0.01	−0.11	0.13	0.87	48.10 (0.19)	0.11	−0.21	0.44	0.49
	Compliance	0.42 (0.06)					47.99 (0.21)				
Weekdays	No compliance	0.43 (0.043)	0.02	−0.09	0.12	0.75	48.08 (0.20)	0.068	−0.24	0.38	0.67
	Compliance	0.41 (0.052)					48.01 (0.20)				
Weekend days	No compliance	0.45 (0.04)	0.08	−0.03	0.19	0.16	48.18 (0.19)	0.39	0.08	0.71	**0.02**
	Compliance	0.37 (0.06)					47.79 (0.22)				
Screen time											
Total week	No compliance	0.46 (0.05)	0.11	−0.00	0.23	0.06	48.14 (0.19)	0.23	−0.09	0.54	0.16
	Compliance	0.35 (0.05)					47.91 (0.21)				
Weekdays	No compliance	0.47 (0.05)	0.10	−0.01	0.20	0.07	48.16 (0.20)	0.20	−0.11	0.50	0.20
	Compliance	0.38 (0.05)					47.96 (0.20)				
Weekend days	No compliance	0.46 (0.04)	0.15	0.03	0.28	**0.02**	48.13 (0.19)	0.28	−0.07	0.62	0.12
	Compliance	0.31 (0.06)					47.85 (0.23)				
Sleep											
Total week	No compliance	0.04 (0.05)	0.03	−0.09	0.16	0.60	48.17 (0.22)	0.20	−0.17	0.56	0.29
	Compliance	0.41 (0.05)					47.98 (0.19)				
Weekdays	No compliance	0.43 (0.06)	0.02	−0.12	0.15	0.81	48.12 (0.23)	0.10	−0.28	0.47	0.62
	Compliance	0.42 (0.05)					48.02 (0.19)				
Weekend days	No compliance	0.45 (0.06)	0.04	−0.10	0.17	0.58	48.17 (0.25)	0.15	−0.27	0.56	0.49
	Compliance	0.43 (0.04)					48.02 (0.19)				
Physical activity and screen time											
Total week	No compliance	0.42 (0.00)	0.07	−0.07	0.20	0.32	48.09 (0.19)	0.24	−0.16	0.63	0.24
	Compliance	0.34 (0.00)					47.85 (0.25)				
Weekdays	No compliance	0.44 (0.04)	0.05	−0.07	0.17	0.38	48.07 (0.19)	0.07	−0.24	0.39	0.66
	Compliance	0.38 (0.06)					48.00 (0.22)				
Weekend days	No compliance	0.44 (0.04)	0.19	0.00	0.37	0.05	48.10 (0.18)	0.46	−0.03	0.95	0.06
	Compliance	0.25 (0.09)					47.63 (0.29)				
Physical activity and sleep											
Total week	No compliance	0.43 (0.04)	0.04	−0.09	0.16	0.57	48.13 (0.19)	0.28	−0.05	0.62	0.10
	Compliance	0.40 (0.06)					48.85 (0.22)				
Weekdays	No compliance	0.44 (0.04)	0.05	−0.07	0.16	0.45	48.15 (0.19)	0.31	−0.01	0.63	0.06
	Compliance	0.39 (0.06)					47.84 (0.21)				
Weekend days	No compliance	0.44 (0.04)	0.07	−0.04	0.19	0.22	48.14 (0.19)	0.35	0.02	0.69	**0.04**
	Compliance	0.37 (0.06)					47.79 (0.23)				
Screen time and sleep											
Total week	No compliance	0.45 (0.04)	0.11	−0.02	0.23	0.09	48.10 (0.19)	0.15	−0.20	0.50	0.40
	Compliance	0.35 (0.06)					47.95 (0.22)				
Weekdays	No compliance	0.46 (0.04)	0.10	−0.02	0.21	0.09	48.09 (0.20)	0.09	−0.22	0.34	0.58
	Compliance	0.36 (0.05)					48.00 (0.20)				
Weekend days	No compliance	0.45 (0.04)	0.15	0.01	0.28	**0.03**	48.10 (0.19)	0.22	−0.15	0.58	0.25
	Compliance	0.31 (0.07)					47.89 (0.23)				
Physical activity, screen time and sleep											
Total week	No compliance	0.44 (0.00)	0.13	−0.05	0.24	0.21	48.09 (0.19)	032	−0.11	0.74	0.14
	Compliance	0.34 (0.08)					48.78 (0.26)				
Weekdays	No compliance	0.44 (0.00)	0.08	−0.06	0.21	0.25	48.09 (0.19)	0.34	−0.01	0..69	0.05
	Compliance	0.36 (0.07)					47.91 (0.22)				
Weekend days	No compliance	0.44 (0.04)	0.16	−0.03	0.35	0.10	48.09 (0.18)	0.40	−0.11	0.90	0.12
	Compliance	0.28 (0.10)					47.69 (0.47)				

All models were adjusted for country, age and educational level of the parent/caregiver who completed the Principal Caregiver’s Questionnaire. Abbreviations: BMI = body mass index, WHR = waist to height ratio, *n* = number, SE = standard error, CI = confidence interval. Significant associations are marked in bold.

**Table 4 ijerph-18-07499-t004:** Association of compliance with the 24 h movement behaviors with BMI z-score and WHR in girls.

		BMI z-Score, Girls *n* = 1164					WHR, Girls *n* = 1164				
Movement Behavior	Compliance/Non-Compliance	Mean (SE)	B	95% CI Lower	95% CI Upper	*p*-Value	Mean (SE)	B	95% CI Lower	95% CI Upper	*p*-Value
Physical activity											
Total week	No compliance	0.27 (0.04)	−0.18	−0.33	−0.03	**0.02**	47.91 (0.18)	−0.39	−0.90	0.12	0.13
	Compliance	0.45 (0.07)					48.30 (0.31)				
Weekdays	No compliance	0.29 (0.04)	−0.08	−0.22	0.05	0.23	47.96 (0.20)	−0.15	−0.60	0.30	0.51
	Compliance	0.37 (0.07)					48.11 (0.26)				
Weekend days	No compliance	0.27 (0.04)	−0.15	−0.27	−0.03	**0.01**	47.95 (0.19)	−0.23	−0.65	0.20	0.29
	Compliance	0.43 (0.06)					48.18 (0.26)				
Screen time											
Total week	No compliance	0.33 (0.05)	0.05	−0.07	0.14	0.41	48.04 (0.21)	0.11	−0.28	0.49	0.59
	Compliance	0.28 (0.05)					47.93 (0.22)				
Weekdays	No compliance	0.35 (0.05)	0.078	−0.05	0.20	0.22	48.17 (0.23)	0.32	−0.09	0.73	0.13
	Compliance	0.27 (0.05)					47.85 (0.21)				
Weekend days	No compliance	0.30 (0.04)	−0.021	−0.13	0.09	0.71	47.99 (0.20)	0.00	−0.38	0.39	0.99
	Compliance	0.32 (0.06)					47.99 (0.24)				
Sleep											
Total week	No compliance	0.40 (0.06)	0.14	0.00	0.27	0.05	48.19 (0.24)	0.30	−0.21	0.80	0.25
	Compliance	0.26 (0.05)					47.89 (0.21)				
Weekdays	No compliance	0.39 (0.06)	0.13	−0.01	0.26	0.07	48.17 (0.25)	0.26	−0.25	0.76	0.32
	Compliance	0.27 (0.04)					47.91 (0.21)				
Weekend days	No compliance	0.46 (0.08)	0.19	−0.04	0.35	**0.02**	48.49 (0.28)	0.62	0.07	1.17	**0.03**
	Compliance	0.27 (0.02)					47.87 (0.20)				
Physical activity and screen time											
Total week	No compliance	0.32 (0.00)	−0.06	−0.26	0.15	0.59	47.99 (0.19)	−0.03	−0.63	0.58	0.93
	Compliance	0.36 (0.11)					48.02 (0.35)				
Weekdays	No compliance	0.31 (0.00)	0.00	−0.16	0.17	0.97	48.01 (0.19)	0.13	−0.35	0.60	0.60
	Compliance	0.30 (0.08)					47.89 (0.27)				
Weekend days	No compliance	0.30 (0.04)	−0.08	−0.28	0.13	0.46	48.00 (0.19)	−0.10	−0.86	0.07	0.79
	Compliance	0.38 (0.10)					48.09 (0.42)				
Physical activity and sleep											
Total week	No compliance	0.30 (0.04)	−0.02	−0.20	0.16	0.84	48.01 (0.18)	0.10	−0.47	0.67	0.73
	Compliance	0.32 (0.09)					47.91 (0.35)				
Weekdays	No compliance	0.31 (0.04)	0.04	−0.12	0.19	0.66	48.02 (0.19)	0.14	−0.36	0.64	0.59
	Compliance	0.28 (0.08)					47.88 (0.29)				
Weekend days	No compliance	0.30 (0.04)	−0.06	−0.20	0.08	0.38	48.01 (0.19)	0.7	−0.40	0.54	0.78
	Compliance	0.36 (0.07)					47.94 (0.30)				
Screen time and sleep											
Total week	No compliance	0.33 (0.05)	0.07	−0.05	0.20	0.24	48.00 (0.20)	0.03	−0.37	0.43	0.89
	Compliance	0.26 (0.05)					47.97 (0.23)				
Weekdays	No compliance	0.36 (0.05)	0.11	−0.00	0.23	0.06	48.14 (0.21)	0.33	−0.05	0.70	0.09
	Compliance	0.24 (0.05)					47.81 (0.21)				
Weekend days	No compliance	0.31 (0.04)	0.01	−0.10	0.13	0.84	48.01 (0.19)	0.09	−0.30	0.49	0.64
	Compliance	0.30 (0.06)					47.92 (0.24)				
Physical activity, screen time and sleep											
Total week	No compliance	0.31 (0.04)	0.08	−0.15	0.33	0.55	48.01 (0.19)	0.28	−0.38	0.94	0.40
	Compliance	0.24 (0.13)					47.73 (0.38)				
Weekdays	No compliance	0.31 (0.04)	0.06	−0.13	0.24	0.55	48.01 (0.19)	0.13	−0.38	0.653	0.61
	Compliance	0.26 (0.09)					47.88 (0.30)				
Weekend days	No compliance	0.31 (0.04)	−0.03	−0.25	0.19	0.78	48.00 (0.19)	0.34	−0.49	0.86	0.94
	Compliance	0.34 (0.11)					47.96 (0.45)				

All models were adjusted for country, age and educational level of the parent/caregiver who completed the Principal Caregiver’s Questionnaire. Abbreviations: BMI = body mass index, WHR = waist to height ratio, *n* = number, SE = standard error, CI = confidence interval. Significant associations are marked in bold.

## Data Availability

Data sharing is applicable upon request.

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
