# Peer review of "Compliance with the 24-Hour Movement Behavior Guidelines and Associations with Adiposity in European Preschoolers: Results from the ToyBox-Study"

_ijerph, 2021, doi:10.3390/ijerph18147499_

Round 1
Reviewer 1 Report
The article is an interesting compliance study of movement behaviours guidelines and association with adiposity in preschool children from different European countries. The manuscript is well written and detailed there however some points in which the manuscript can be improved
1.- the study was conducted at a specific time of the year? Outdoor activity versus indoor activity.
2.- the compliance study of each country can be added in the supplementary files.
3.- is there any information on the calorie intake related to movement?
4.- physical activity guidelines during the week are dependent on the kindergarten and can be measured more accurately than the weekend. Do the authors have information on the amount of time that each centre devotes to physical education each week?
5.- it would be very important that the authors would comment on possible changes in the guidelines to enhance compliance.
Author Response
Please see the attachment. Answers to your remarks can be found below "REVIEWER 1".

Reviewer 2 Report
Please confirm the attached file.

Author Response
Please see the attachment. Answers to your remarks can be found below "REVIEWER 2" on page 3.
